# An Exploratory Case Study of Software Project Management under Large Language Model Usage

## Abstract

This paper investigates how software project managers perceive changes in their work and learning demands associated with the use of large language models (LLMs) in software projects. Motivated by a gap in software engineering research, which has largely focused on task-level and developer-centered uses of LLMs, we adopted an exploratory case study approach to capture managerial perspectives on LLM usage in practice. Based on the experience of software project managers working in a large, multi-project software organization, our analysis indicates that LLM usage is perceived as influencing planning, estimation, coordination, monitoring, and governance activities rather than introducing new formal management practices. Participants described LLMs as becoming embedded in everyday project work, with uneven effects on productivity and increased attention to review, validation, and responsibility. Learning demands were perceived as experiential and incremental, centered on understanding LLM capabilities and limitations, critically assessing generated artifacts, and guiding responsible use within teams. Our findings provide preliminary empirical evidence on how software project management work is perceived and adapted in contexts where LLMs are integrated into ongoing software development practice.

## Keywords

software project management, software managers, LLMs, case study

### ACM Reference Format:

Anonymous Author(s). 2018. An Exploratory Case Study of Software Project Management under Large Language Model Usage. In *Companion Proceedings of the 34th ACM Symposium on the Foundations of Software Engineering (FSE '26), July 5 - 9, 2026 Montreal, Canada.* ACM, New York, NY, USA, 9 pages. https://doi.org/XXXXXXX.XXXXXXX

## 1 Introduction

Software project management is a central activity in software development, concerned with planning, coordinating, and guiding work toward the delivery of software systems within organizational and project constraints [13, 21]. The literature characterizes software project management as a socio-technical practice that goes beyond scheduling and budgeting, encompassing coordination across roles, communication, and alignment between technical work and organizational objectives [13, 19, 21]. Prior studies indicate that project

management in software development is enacted through situated judgment and adaptation rather than strict adherence to prescribed processes, including in agile contexts where formal management structures are reduced [10, 15]. As a result, project management activities remain embedded in everyday development work across various development approaches [10, 15].

In this context, the software project manager role is described as coordinative and interpretive, involving mediation between organizational constraints, stakeholder expectations, and development work [21, 29]. The literature associates this role with managing cost, schedule, scope, and quality considerations, as well as supporting communication and coordination across team members and roles [10, 19]. Software project success is often understood by managers in ways that extend beyond time and budget adherence, including meeting user requirements, delivering working systems, and achieving perceived quality in the delivered outcomes [21]. These perspectives suggest that software project managers influence how development practices are organized and evaluated throughout a project [21, 29].

More recently, the literature reports an increasing presence of artificial intelligence (AI) and large language models (LLMs) in software development and related management activities, including coding, documentation, testing, learning, estimation, monitoring, and coordination [2, 12, 23]. These technologies are framed as assistive, supporting routine and information-intensive tasks while retaining human oversight and decision-making responsibility [2, 5]. At the same time, existing work primarily emphasizes how software practitioners adapt their development routines to AI and LLM usage, with comparatively limited attention to how software project managers adapt their practices, responsibilities, and skills in response to these changes [12, 23]. While the literature notes that managers are beginning to use AI-based tools for planning, reporting, and coordination activities, it provides limited insight into what software project managers need to learn or change to support teams that increasingly rely on LLMs [2]. In response, this study focuses on software project management in the context of AI and LLM adoption and poses the following research question (RQ):

> **RQ1 – Perceived Effects of LLMs in Software Project Management**
>
> *How do software project managers perceive the changes and learning demands introduced by LLMs in software project management?*

To answer our RQ, we conducted an exploratory case study in a large software organization. We focused on the software project manager role as the unit of analysis and collected data through an online qualitative questionnaire administered to ten managers responsible for planning and coordinating ongoing projects within the organization. The instrument included open-ended questions

designed to capture perceptions of managerial practices, challenges, and adaptation processes related to LLM usage. Responses were analyzed using thematic analysis complemented by descriptive statistics to support contextual interpretation. Our findings suggest that LLM usage is perceived as shaping planning, estimation, coordination, monitoring, and governance activities rather than creating entirely new management practices. Participants described LLMs as increasingly integrated into everyday project work, producing uneven productivity effects and requiring greater attention to review, validation, and accountability. Learning demands were characterized as experiential and incremental, focusing on developing an understanding of LLM capabilities and limitations, critically evaluating generated artifacts, and supporting responsible use within teams.

This paper makes the following contributions:

- **An analysis of how LLM adoption reshapes managerial work**, demonstrating that changes occur mainly through increased reliance on managerial judgment in planning, coordination, and risk management, rather than through the introduction of new formal project management practices.
- **An identification of emerging learning and governance demands**, highlighting how managers incrementally develop capabilities to evaluate LLM-assisted outputs, guide responsible tool use, and handle accountability and governance through case-by-case managerial decisions rather than formalized organizational frameworks.

Based on this introduction, the remainder of the paper is organized as follows. Section 2 presents the background of this study. Section 3 describes our methodology. Section 4 reports our findings, which are discussed in Section 5. Section 6 discusses threats to validity. Finally, Section 7 summarizes our main contributions and final considerations.

> **Data availability statement:** To ensure verifiability and replicability, we provide the replication package of this study available at https://figshare.com/s/9e721bea3d114f7b9a24.

## 2 Background

This section situates software project management and the use of LLMs within longer trajectories of change in software engineering practice. The focus is on how skills, expectations, and task boundaries have evolved, providing context for understanding contemporary management challenges in AI-supported development environments.

### 2.1 The Role of Software Managers

Early software engineering literature describes software managers as responsible for organizing work in settings characterized by technical complexity, limited automation, and high delivery risk [13]. Managerial skill during this period is associated with planning, estimation, progress tracking, and quality control, reflecting a reliance on formal coordination mechanisms and documentation to manage uncertainty [13, 21]. Accounts from industrial settings also emphasize the importance of managerial judgment in balancing competing

constraints and responding to deviations from plan, particularly in large and long-running projects [21].

As development practices diversified, the literature began to describe software management as increasingly adaptive. Work on development practice shows that formal processes often diverge from how work is actually carried out, requiring managers to rely on experience, negotiation, and informal coordination to address evolving conditions [15]. Research on agile and hybrid environments reports that management responsibilities persist even when formal roles are minimized, with managers supporting prioritization, alignment, and coordination across teams and stakeholders [4, 10]. In this context, managerial skill is framed less in terms of enforcing predefined procedures and more in terms of maintaining shared understanding and managing interdependencies across the software team [19].

More recent literature places greater emphasis on the socio-technical aspects of software management. Software managers are described as operating across organizational, technical, and social boundaries, requiring skills in communication, interpretation, and decision-making under partial information [29]. Current practices associate managerial effectiveness with the ability to interpret software project dynamics, mediate expectations, and support team-level decision making rather than direct technical control [4]. Educational perspectives further indicate that these capabilities require deliberate development beyond technical training, particularly in coordination, accountability, and organizational awareness [20].

### 2.2 LLMs in Software Engineering Practice

LLMs are increasingly present in the software engineering environment, with use extending across multiple roles and stages of development [12]. Recent studies report that developers, testers, analysts, and other practitioners incorporate LLMs into activities such as requirements-related work, implementation, debugging, documentation, testing support, and learning [12, 16, 27, 32]. Rather than being tied to a specific phase or artifact, LLMs are used opportunistically as flexible resources that practitioners draw on in response to task demands and situational needs [6, 17, 18]. This pattern reflects an environment in which interaction with LLMs becomes part of everyday software work across roles.

Across development activities, the availability of LLMs appears to influence how practitioners engage with software tasks. Instead of replacing established practices, LLMs are used to support exploration, clarification, and intermediate production, with responsibility for interpretation and final decisions remaining with humans [7, 25]. Evidence from collaborative and team-based settings shows that LLMs are often used early to explore design options or generate initial solutions, while humans retain responsibility for integration, architectural alignment, and validation [22, 26].

LLM use also extends beyond developer-facing activities into tasks related to coordination and management. Currently, practitioners and managers are using LLMs to assist with estimation, documentation synthesis, reporting, monitoring, and aggregation of information across project artifacts [1, 2, 5, 30]. This broader uptake suggests a changing software engineering environment in which multiple roles interact with AI-supported tools. At the same time, existing accounts provide limited detail on how this

shift affects managerial responsibilities, decision boundaries, or skill development, as most discussions remain focused on task-level support rather than changes in management practice [2, 23].

## 3 Methodology

We adopted a case study research design in accordance with established software engineering guidelines for empirical case studies [9, 24]. In software engineering, a case study is an empirical approach that investigates a contemporary phenomenon in its real-world context. This design is appropriate for studying socio-technical phenomena that are enacted through everyday practice, interpretation, and coordination rather than through controlled interventions. We designed this study as an exploratory case study to investigate how software project managers perceive changes in their work and learning demands in response to the increasing use of LLMs in software projects. The study does not seek to evaluate a specific tool, compare alternatives, or establish causal relationships. The focus is on understanding managerial sense-making, adaptation, and perceived challenges in a natural organizational setting. This positioning aligns with prior case study research in software engineering that emphasizes contextualization and theoretical insight rather than statistical generalization [9].

### 3.1 Case Study Context

The case concerns a large, long-established software and innovation organization operating primarily in South America, with additional presence in North America and Europe. The organization functions as a research, development, and innovation center and delivers software-intensive projects for industrial clients across multiple sectors. The workforce includes software developers, quality assurance professionals, designers, data scientists, automation specialists, and professionals in software and technical management roles. At the time of the study, the organization was running at least 70 software projects concurrently. These projects covered several domains, including finance, healthcare, education, digital platforms, industrial systems, and data-driven services. Project work was organized around multidisciplinary teams, with management activities embedded in planning, coordination, monitoring, and decision-making across projects. We selected this organization as the case because it provides an analytically rich setting for studying software project management under conditions of increasing LLM adoption. The scale of operations, the diversity of domains, and the coexistence of many concurrent projects create conditions in which managerial practices, technical work, organizational constraints, and AI-mediated development are closely intertwined. These characteristics make the case suitable for an exploratory software engineering case study focused on socio-technical change.

### 3.2 Participant Selection

We defined the unit of analysis as the software project manager role. Within the organization, fifty-seven individuals identified their primary role as software project manager or technical manager, being responsible for planning, coordinating, and supervising software projects and teams. From this bounded population, we applied random sampling and contacted twenty-five managers to invite them to participate in the study. Participation was voluntary, and ten managers completed the questionnaire.

Although the number of respondents is limited, these participants were collectively responsible for at least 30 of the 70 projects active in the organization at the time of data collection. This sampling strategy sought to balance feasibility with analytical relevance. Rather than aiming for statistical representativeness, the study prioritized access to practitioners directly involved in managing projects influenced by LLM adoption. Consequently, the resulting sample constitutes a self-selected qualitative sample suitable for exploratory case study research in software engineering, where contextual insight and information richness are central concerns.

### 3.3 Pilot

We validated the questionnaire through a pilot study with two researchers experienced in software management research. Feedback from the pilot led to improvements in wording clarity, refinement of concepts related to LLM capabilities and limitations reported in prior work, and a reduction in overlap among questions addressing planning, learning, and governance. Table 1 presents the full instrument. Data collection happened during January 2026.

### 3.4 Data Collection

We initially planned to collect data through semi-structured interviews, which are commonly used in software engineering research to capture practitioners' experiences and interpretations of managerial work and decision-making. During recruitment, we observed that this approach was impractical given the professional profile of the targeted participants. Managers reported highly restricted availability, which limited the feasibility of synchronous interviews. We therefore transformed the interview guide into an online qualitative questionnaire, following prior software engineering research that adapts qualitative instruments to accommodate participants' time constraints while preserving open-ended reflection. This adaptation allowed participants to respond asynchronously while preserving the reflective and open-ended nature of the original questions. The questionnaire was designed to elicit detailed accounts of managerial practices, perceived changes, and learning demands associated with LLM usage in software projects. The questions were developed based on prior work on LLM usage in software engineering and on emerging discussions of AI-supported activities in software project management [2, 12, 23].

### 3.5 Data Analysis

The questionnaire yielded both qualitative and quantitative data. We analyzed open-ended responses using thematic analysis [31], while we summarized demographic and contextual data using descriptive statistics [11]. We conducted thematic analysis following procedures commonly recommended for qualitative research in software engineering [28]. We coded the received answers following iterative cycles. In the first cycle, we read each response carefully and assigned codes capturing the core idea expressed by the participant. In the second cycle, we grouped conceptually related codes into broader categories representing recurring perceptions, challenges, and learning demands. We documented a rationale for each

**Table 1: Survey Questionnaire**

---

**LLMs AND SOFTWARE MANAGEMENT**

1. Considering software projects in which large language models (LLMs) such as ChatGPT or GitHub Copilot are used by technical roles, which skills are most important for planning and estimation in this context?
2. How have you developed or modified your management skills due to the use or potential use of LLMs in software teams?
3. Which control and monitoring skills have you needed to develop or strengthen due to LLM usage in software development?
4. Which challenges best reflect your experience with control and monitoring in projects involving LLMs? (Estimation and planning under uncertainty; progress and productivity monitoring; coordination between human work and AI tools; decision making under incomplete or variable information; risk management related to LLM usage; traditional skills remain sufficient; other.)

---

**LEARNING AND LLMs**

5. Which new skills have you needed to learn in practice to manage LLM mediated software projects?
6. Which area has presented the greatest difficulty from a managerial skills perspective? (Technical understanding of LLM capabilities and limitations; quality assurance and validation; risk management and decision making under uncertainty; leadership and coordination of AI augmented teams; governance, ethics, and responsibility; continuous learning and professional adaptation.)

---

**GOVERNANCE**

7. What types of decisions related to governance, quality, or responsibility regarding LLM usage have you needed to make in the projects you manage?
8. Which aspect of governance has required the greatest managerial effort due to LLM usage? (Defining limits for use; ensuring quality and reliability of artifacts; authorship and responsibility; compliance with organizational or legal policies; communication and expectation alignment; no additional effort.)
9. How are decisions related to LLM usage currently handled in your projects? (Case by case; organizational guidelines adapted to the project; discussed within the team; defined by external areas such as legal or security; no consistent approach; other.)

---

**ABOUT YOU**

10. How long have you worked in management or leadership roles in software projects? (Less than 2 years; 2–5 years; 6–10 years; more than 10 years.)
11. How many software projects do you currently manage or supervise simultaneously? (1 project; 2–3 projects; 4–5 projects; more than 5 projects.)
12. Approximately how many people are under your direct or indirect responsibility? (Up to 5; 6–10; 11–20; more than 20.)
13. How frequently do team members use LLMs as part of daily work? (Never; rarely; sometimes; frequently; always.)
14. In which activities are LLMs most used in your projects? (Code writing or review; test generation; documentation support; design or architecture exploration; requirements clarification; technical problem solving; not used; other.)
15. How do you describe your gender identity? (Woman; man; non binary; another gender identity; prefer not to inform.)

---

category to clarify its meaning and analytical scope. A final refinement cycle focused on ensuring conceptual clarity and consistency across themes. One researcher conducted the initial coding. A second researcher reviewed the codes and categories. We discussed the disagreements until we reached a shared interpretation. This process was used to enhance credibility, transparency, and reflexivity, and to reduce the influence of individual researcher assumptions. Although the primary emphasis of the analysis was qualitative, we used descriptive statistics to summarize participant characteristics and to indicate the frequency of selected response options and themes. These summaries support contextual interpretation rather than inferential claims.

## 3.6 Ethics

We conducted the study in accordance with institutional guidelines for research involving human participants. We obtained informed consent electronically at the beginning of the questionnaire. We informed participants about the study objectives, voluntary participation, anonymity, and their right to withdraw before submission. We did not collect personally identifiable information. We analyzed all data in aggregated form and used them exclusively for academic research purposes.

## 4 Results

Our sample comprised ten participants with heterogeneous managerial backgrounds and responsibilities (see Table 2). With respect to gender identity, four participants identified as women, five as men, and one preferred not to disclose. In terms of managerial experience, four participants reported two to five years in management or leadership roles on software projects, three reported six to ten years, and three reported more than ten years. Workload varied across participants: six managed or supervised two to three projects simultaneously, two oversaw four to five projects, and two were responsible for a single project. Responsibility scope was substantial, as five participants oversaw more than twenty people, two supervised between eleven and twenty people, and three were responsible for teams of six to ten people.

**Table 2: Participant Demographics and Responsibilities**

| Characteristic | Distribution |
|---|---|
| Gender identity | 4 women; 5 men; 1 undisclosed |
| Managerial experience | 4 (2–5 years); 3 (6–10 years); 3 (>10 years) |
| Projects supervised | 2 (1 project) 6 (2–3 projects); 2 (4–5 projects); |
| People supervised | 3 (6–10 people) 2 (11–20 people); 5 (>20 people); |

LLM usage was well established across participants' projects. Five participants reported that team members always use LLMs as part of daily work, four reported frequent use, and one reported occasional use. The reported uses of LLMs concentrated on a small set of recurring activities. *Test generation* was the most frequently reported activity, mentioned by eight participants, followed by *code writing or code review* in seven responses. *Investigation and resolution of technical problems* appeared in six responses, while *support for technical documentation* was reported by five participants. *Exploration of design or architectural alternatives* and *clarification or refinement of requirements* were each mentioned by three participants, and *experimental code generation* was reported by one participant. Overall, these patterns suggest that LLM usage is primarily embedded in core development and quality-related activities, with more limited use in design exploration, requirements-related work, and experimental code generation.

**📊 Overview: LLM usage across participants' projects**

**Frequency of LLM usage in projects**

| Usage frequency | Participants |
| --- | --- |
| Always used | 5 |
| Frequently used | 4 |
| Occasionally used | 1 |

**Main activities supported by LLMs**

| Activity | Mentions |
| --- | --- |
| Test generation | 8 |
| Code writing/review | 7 |
| Technical problem resolution | 6 |
| Documentation support | 5 |
| Design exploration | 3 |
| Requirements clarification | 3 |
| Experimental code generation | 1 |

Overall, LLM usage is embedded mainly in development and quality-related tasks, with more limited adoption in design and requirements activities.

## 4.1 Perceived Changes in Software Project Management Practices with LLM Usage

Across participants, planning and estimation in projects where LLMs are used were described as increasingly dependent on managerial judgment, contextual awareness, and caution rather than on the adoption of new formal estimation techniques. Participants emphasized that effective planning requires understanding team maturity, familiarity with the product domain, and awareness of how LLM usage varies across roles and tasks. Several participants highlighted the importance of understanding LLMs' limitations to avoid unrealistic assumptions about productivity. One participant stressed that managers need to recognize that *"the LLM will not solve all problems and should not be applied in all contexts"* (P006), while another emphasized the need to avoid *"unrealistic productivity expectations"* by understanding what these tools do well and where they fall short (P003).

Communication, engagement, and alignment across roles were also a central element in participants' narratives about planning and estimation. Rather than treating estimation as a purely technical activity, participants described it as closely tied to coordination and shared understanding within the team. One participant emphasized that *"management and engagement skills are essential"*, alongside *"good communication to bring and share initiatives related to LLM usage with the team"*, including providing time for learning and monitoring outcomes through associated metrics (P001). One participant further noted that the impact of LLMs is uneven across tasks, stating that *"while some tasks may be accelerated, others start to require more time for review and validation"* (P003). This variability was described as complicating effort estimation and reinforcing the need for flexible planning approaches that can be adjusted as project conditions evolve.

At the same time, continuity with established planning practices was evident. Some participants reported continuing to rely on standard estimation approaches, particularly given the relatively recent adoption of LLMs, while observing gradual improvements in delivery speed among developers who actively use these tools. As one participant explained, *"I still use the standard estimation, but in practice I have seen delivery time gradually improve"* (P005). Others explicitly stated that the same skills valued prior to LLM adoption remain relevant. One participant emphasized the continued importance of *"the same skills already established before the LLM era, such as knowledge of engineering process methodologies used in the project, critical thinking, problem solving, experience with project management in the technologies used, and, when possible, knowledge of the application domain"* (P008). However, participants also pointed to differentiated effects across experience levels, noting that less experienced developers may struggle to assess output quality. One participant cautioned that *"people with lower seniority may not have the experience to identify hallucinations or assess the quality of LLM outputs"* (P009), introducing additional uncertainty into planning and estimation decisions.

As planning and estimation became more judgment-driven and less predictable, participants described corresponding shifts in how they approached control, coordination, and risk during project execution. *Coordination between human work and AI-assisted tools* was reported by four participants, reflecting increased effort to align individual uses of LLMs with collective workflows and shared project goals. In parallel, *risk management associated with LLM usage* was also reported by four participants, driven by concerns related to quality, reliability, dependence on tools, and potential legal or ethical implications. Challenges related to *monitoring progress and productivity* were reported by three participants, who attributed these difficulties to reduced visibility into how work is performed when tasks are mediated through individual interactions with LLMs. A smaller number of participants, specifically two, highlighted increased *decision making under uncertainty*, attributed to non-deterministic outputs, variable results, and limited explainability. At the same time, four participants indicated that *traditional managerial skills remain sufficient*, suggesting that these changes are experienced as selective adjustments to established practices rather than as a comprehensive transformation of software project management.

## 4.2 Learning Demands and Skill Adaptation in LLM-Mediated Software Project Management

Across participants' experiences, learning was described as situated and experience-driven, emerging from day-to-day engagement with LLMs rather than from formal training. Software managers emphasized the need to adapt incrementally, often by first understanding the tools themselves and then translating that understanding into managerial oversight. They reported that learning began with **understanding what LLMs are and where they can be applied in practice**, particularly across different project activities and roles. As one individual explained, learning involved *"understand[ing] what LLMs were about and know[ing] where and how the team started to apply them in projects"* (P001), while another noted the need to *"understand how to use the tools in order to be able to follow the team"* (P005). A closely learning demand was related to **learning through**

**direct, hands-on use of LLMs**. Software managers described personally using LLMs as a prerequisite for effective supervision and interpretation of team practices. One participant stated that *"the first step was to use the tools myself"*, explaining that this experience made it easier to understand both gains and difficulties faced by the team (P006). For these individuals, learning was experiential and exploratory, enabling adaptation of monitoring and coordination practices based on first-hand understanding.

Participants also highlighted the need to develop **critical evaluation of LLM-generated outputs**. Learning to assess quality, reliability, and contextual appropriateness was described as the key to managing LLM-mediated work. One software manager explained the need to learn how to *"evaluate critically the results generated by the tools, understanding when they really help and when they can introduce errors or inconsistencies"* (P003). Others emphasized that this learning was necessary to prevent the erosion of review discipline, particularly given the volume and convenience of the artifacts generated. Similarly, individuals reported learning how to strengthen **the review, validation, and integration of LLM-assisted artifacts**. This included reinforcing testing and validation practices and ensuring that LLM-generated outputs could be coherently integrated into existing workflows. One participant noted the importance of coaching teams to avoid losing *"important skills for reviewing and critically assessing generated code"* (P008), while another described learning to explicitly account for additional effort required for review and validation when managing projects (P003).

Participants also described learning to **adjust planning and estimation assumptions based on empirical experience**. Rather than assuming automatic productivity gains, software managers reported learning to revise estimates based on observed outcomes. One participant explained that they learned to stop assuming automatic time savings and to consider *"the additional effort for review, validation, and integration"* (P003), while another noted that although standard estimation was still used, *"in practice I have seen delivery time gradually improve"* as experience accumulated (P005). Learning demands also extended to **communicating expectations and guiding responsible use**. Participants described the need to learn how to align team expectations, balance productivity with responsibility, and reinforce appropriate use of LLMs. One software manager emphasized strengthening communication to *"align expectations about the responsible use of LLMs"* (P003), while another described their role as reinforcing use and *"show[ing] the positive results of using the tools within the team"* (P005).

In some cases, learning involved acquiring **operational and technical skills related to LLM usage**. These included prompt writing, preparing specifications for code generation, and configuring tools to support automated review. One participant explicitly reported learning *"prompt writing and specification files for code generation, [and] configuration of tools for automated review"* as part of managing LLM-mediated projects (P008). Related to prompting, managers reported learning about **legal, ethical, and confidentiality constraints** that are related to LLM adoption. For some individuals, this learning defined the boundaries of acceptable use rather than expanding managerial practice. One participant referred to *"legal issues regarding how far we could use AI"* (P004), while another explained that *"due to confidentiality issues in the projects, the use of AI is limited"* (P009).

Finally, we had software managers who emphasized that learning about LLM usage in their projects did not replace existing competencies but involved **reinforcing and extending traditional managerial skills**. These participants described continuity with prior practice, noting that foundational managerial skills remained essential and, in some cases, became more critical. One individual stated that *"the fundamental skills remain the same, if not even more required"* (P008), while another acknowledged the need to invest in learning over time while *"for now, [continuing] with the traditional skills"* (P010).

## 4.3 Governance, Responsibility, and Managerial Decision Making in LLM-Mediated Projects

From a managerial skills perspective, reported difficulties related to LLM use were concentrated on governance, technical understanding, and uncertainty management rather than on a single dominant challenge. *Governance, ethics, and responsibility* was the most frequently reported area of difficulty, mentioned by six participants, reflecting recurring uncertainties regarding data use, artifact authorship, compliance, and accountability in cases of failure. *Technical understanding of LLM capabilities and limitations* followed, reported by five participants, who described difficulties in assessing whether LLM-generated outputs were adequate, reliable, or appropriate for the project context. *Risk management and decision making under uncertainty* was reported by four participants, who noted the need to make decisions even when the impact of LLM use on deadlines, quality, or rework was unpredictable. *Leadership and coordination of AI-augmented teams* and *continuous learning and professional adaptation* were each reported by three participants, indicating challenges in defining roles, responsibilities, and expectations, as well as in updating practices in the absence of consolidated guidance or formal training. Finally, *quality assurance and validation of LLM-assisted work* was reported by three participants, indicating persistent uncertainty about how to review, test, or validate artifacts produced with LLM support.

When participants were asked to reflect specifically on governance aspects that demanded greater managerial effort, responses further clarified how these difficulties were experienced in practice. *Ensuring the quality and reliability of generated artifacts* was the most frequently reported governance concern, mentioned by seven participants, reflecting challenges related to the opacity of artifact origins and varying degrees of dependence on LLMs. *Communicating and aligning expectations with the team* was reported by five participants, highlighting the need to manage differing interpretations across roles regarding acceptable LLM usage. *Defining limits for LLM usage* was reported by four participants, pointing to uncertainty about which activities should or should not be supported by these tools. *Ensuring compliance with organizational or legal policies* was reported by three participants, indicating that existing rules do not always explicitly address LLM-mediated work. *Authorship and responsibility attribution* was reported by two participants, reflecting difficulties in assigning responsibility for decisions or failures when software artifacts are produced with LLM assistance. In contrast, one participant indicated that *none of these governance aspects generated significant additional effort*, suggesting that governance implications are not experienced uniformly across projects.

These findings indicate that governance and responsibility in LLM-mediated software projects are primarily enacted through managerial judgment, expectation alignment, and quality-focused oversight rather than through fully formalized governance mechanisms. Several software managers described making decisions about acceptable use, limits, and accountability on a case-by-case basis, often centered on reinforcing human responsibility for LLM-assisted outcomes. One participant explained the need to define *"in which activities LLMs could be used freely and in which cases their use would require more rigorous review, especially in critical code, requirements, and architectural decisions"*, while emphasizing that *"any result generated by LLMs continued to be the responsibility of the team, not the tool"* (P003). Others described governance efforts in terms of reinforcing review practices, such as *"reinforcing that technical profiles need to review what the AI is suggesting"* (P005), deciding *"what data we could give to the AI and whether people were correcting the code coming from the AI"* (P004), and ensuring *"human review of generated code and guidance on when to avoid prioritizing LLM use for problems where official documentation is more effective"* (P008). In addition, some managers highlighted the importance of *"aligning expectations with stakeholders and ensuring transparency about where and how LLMs were being used within the project"* (P003), while at least one participant reported that *"so far, I have not needed to make specific decisions related to the use of LLMs in the projects I manage"* (P010), indicating that governance demands vary across contexts.

## 4.4 Answering RQ$_1$: How Do Software Project Managers Perceive the Changes and Learning Demands Introduced by LLMs?

Software project managers perceive the introduction of LLMs as producing selective and uneven changes in software project management practices, alongside new and intensified learning demands, rather than as a comprehensive transformation of their role. Managers described planning and estimation as becoming more judgment-driven and context-sensitive, with greater caution required in interpreting productivity gains and increased attention to coordination, visibility, and risk during project execution. At the same time, several managers reported continuity with established practices and reliance on traditional managerial skills.

Learning demands were perceived as experiential and incremental, centered on developing technical understanding of LLM capabilities and limitations, critically evaluating and validating LLM-assisted artifacts, and guiding responsible use within teams, while largely extending rather than replacing existing managerial competencies. Finally, managers characterized governance and responsibility related to LLM usage as requiring ongoing managerial judgment, expectation alignment, and reinforcement of human accountability, typically handled on a case-by-case basis rather than through formalized governance frameworks.

---

### 📊 RQ$_1$ – Perceived Effects of LLMs in Software Project Management

**Perceived changes in project management**

| Reported perception |
| --- |
| Selective and uneven changes rather than a full transformation of managerial roles |
| Greater reliance on managerial judgment and contextual interpretation of productivity gains |
| Increased attention to coordination, visibility, and risk monitoring |
| Many established management practices and skills remain central |

**Emerging learning demands**

| Managerial response |
| --- |
| Incremental and experience-based learning about LLM capabilities and limitations |
| Need to critically evaluate LLM-generated artifacts |
| Managers support responsible adoption rather than replace existing competencies |

**Governance implications**

| Observed pattern |
| --- |
| Responsibility management handled through managerial judgment and expectation alignment |
| Governance typically informal and addressed case by case rather than through structured frameworks |

---

## 5 Discussion

**Comparison with the Literature.** Prior software engineering research on LLMs has largely concentrated on task-level use, including code generation, debugging support, documentation, testing assistance, and learning activities [12, 16, 27, 32]. More recent work has started to report collaborative and team-based uses, often characterizing LLMs as supporting exploration, clarification, and intermediate artifact production, while responsibility for interpretation, validation, and integration remains with humans [7, 22, 26]. Our findings are consistent with this body of work with respect to usage patterns and underlying mechanisms. Our participants described LLMs as flexible tools embedded in everyday development and quality-related work, accelerating certain activities while introducing additional effort for review, validation, and contextual assessment.

The contribution of this study lies in its focus on software project management rather than on individual or task-specific perspectives. Existing studies occasionally note managerial or coordination-related uses of LLMs, but these aspects are typically treated as secondary [2, 5]. In contrast, our findings indicate that managers explicitly perceive LLM adoption as influencing planning, estimation, coordination, and governance practices. Rather than prompting the adoption of new formal management techniques, LLM usage appears to increase reliance on managerial judgment, contextual awareness, and experience, particularly when dealing with uneven

productivity effects, reduced visibility into work processes, and non-deterministic outputs. This observation is consistent with earlier findings about software management that emphasized the need for interpretive work, negotiation, and decision-making under partial information [21, 29].

Our findings also extend prior research by foregrounding learning demands and governance as key managerial concerns nowadays. While much of the existing literature focused on individual skills such as prompt writing or tool proficiency, software managers described learning as experiential, incremental, and closely connected to oversight responsibilities, including quality assurance, expectation alignment, and responsibility attribution. In this sense, our results align with socio-technical accounts of software management that emphasize coordination, accountability, and uncertainty management [4, 19]. Overall, our study suggests that managers perceive LLMs as tools that become embedded in existing software project management work, influencing how planning, coordination, oversight, and governance are carried out in practice. Rather than redefining the management role, LLMs are described as affecting day-to-day managerial activities by altering task visibility, effort distribution, and review demands, and by requiring managers to actively guide, monitor, and constrain their use within ongoing projects.

**Implications for Research and Practice.** From a research perspective, this study contributes to the study of AI in software engineering by shifting attention from task-level outcomes to managerial practice. Our findings suggest that understanding the impact of LLMs requires greater focus on how software project managers adapt their planning, coordination, and governance activities in AI-mediated work. Future research may investigate how organizational context, project criticality, or regulatory constraints influence managerial responses to the use of LLMs. Longitudinal studies may further characterize how learning demands and governance practices change as LLM use becomes more established, while comparative studies may investigate whether similar patterns are observed across domains or organizational settings.

From a practice perspective, our findings indicate that LLM adoption does not diminish the importance of established managerial skills, rather, it places greater emphasis on judgment, communication, and oversight. Participants in this study did not report relying on new formal planning or governance approaches. Instead, they described adapting existing practices to account for uncertainty, uneven task impacts, and additional effort related to review and validation. This suggests that software organizations should be cautious about assuming uniform productivity gains from LLM adoption and should support managers in developing contextual understanding, critical assessment skills, and mechanisms for aligning expectations within teams. Training and guidance may therefore be more effective when oriented toward responsible use and coordination rather than toward technical instruction alone.

## 6 Threats to Validity

As with any qualitative case study, our study has limitations inherent to the method [9, 24]. First, our findings are based on the experience of 10 software project managers. While this number may be questioned from a quantitative perspective, such criticism is not valid for qualitative case studies, which prioritize depth of experience over sample size [9]. Prior studies in software engineering and related fields report credible exploratory work with similar or smaller samples, including studies with as few as six participants [3, 8]. In our case, the 10 participants collectively managed approximately 30 active software projects, representing about 42% of all projects running in the company at the time of data collection, providing broad exposure to ongoing project practices within the organization. Second, our study does not seek statistical generalization. Instead, we aim for transferability [9]. To support this, we provide detailed information on the organizational context, participants, and observed software project management practices, enabling other researchers and practitioners to assess how our findings apply to their own settings. Third, our analysis relies on self-reported accounts obtained through interviews, which reflect participants' perceptions rather than direct observation. This is a recognized characteristic of case studies and does not constitute a validity weakness when findings are grounded in recurring patterns and supported by quotations. Finally, common criticisms such as the absence of quantitative data, limited reproducibility, or lack of causal claims are not applicable, as our study is exploratory and should be evaluated using qualitative criteria such as credibility, rigor, and transferability [9].

## 7 Conclusions and Future Work

Our study investigated how software project managers perceive changes in their work and learning demands associated with the use of LLMs in software projects. Using an exploratory case study conducted in a large, multi-project software organization, we examined managerial sense-making, adaptation, and responsibility in contexts in which LLMs are embedded in everyday development and coordination activities. Rather than focusing on specific tools or evaluating technical performance, our study aimed to understand how managers interpret and respond to the use of LLMs within ongoing software project management practice. Our findings indicate that managers perceive LLMs as becoming embedded in existing project management work rather than as introducing new roles or formal management approaches. LLM adoption does not reduce the importance of management, nor does it automate managerial work. Instead, it alters the conditions under which management is performed, increasing attention to uncertainty, coordination, learning, and accountability while remaining embedded in established project management practices. Overall, our study provides empirical insight into how software project managers perceive and adapt to the use of LLMs in practice. For future work, we intend to expand the body of knowledge generated in this study by administering a survey to software project managers to assess the prevalence and variation of the identified patterns across a broader range of organizational contexts. In addition, we plan to develop training materials for software managers based on the practices, learning needs, and governance challenges identified in this study, thereby supporting more informed and responsible use of LLMs in software project management.

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
