# OpenReview forum: "An Exploratory Case Study of Software Project Management under Large Language Model Usage"
_ACM.org/AIWare/2026/Conference — Submitted to AIware 2026_

### Official Review · Reviewer_Dfoo · 2026-03-08

**Rating:** 3
**Confidence:** 4

**Review:**

Strengths

1. I like the perspective of the paper. A lot of current work on LLMs in software engineering stays at the level of coding, testing, or developer productivity, while this paper looks at software project managers. That angle is useful and feels underexplored.

2. The paper is also reasonably clear about its study design. The case context, participant selection, pilot, questionnaire design, thematic analysis process, and threats to validity are all described in a fairly transparent way, which helps the reader understand what this study can and cannot support.


Weaknesses

1. My main concern is that the empirical base is still quite limited. The study is built on ten managers from a single organization, so the findings are interesting, but still feel preliminary. It is hard to know how much of the result is specific to this company and how much would carry over to other contexts.

**Summary:**

The paper reports an exploratory case study on how software project managers perceive the impact of LLM usage in software projects. Instead of focusing on developers or task-level uses of LLMs, it looks at managerial work such as planning, estimation, coordination, monitoring, and governance. The study is based on responses from ten project managers in a large software organization, and the main takeaway is that LLMs do not seem to create entirely new management practices, but they do make managerial work more judgment-driven, with more attention needed for review, validation, responsibility, and expectation alignment.

---

> ### Author Response · Authors · 2026-03-13
>
> Dear Reviewer,
>
> Thank you for your thoughtful and supportive review. We are particularly grateful for your comment about the focus on software project managers. As you noted, much of the current research on LLMs in software engineering concentrates on coding, testing, or developer productivity. One of our main motivations in conducting this study was to explore a perspective that is still relatively underexamined in the literature, not just consider LLMs, but in many other contexts.
>
> Regarding the empirical scope, we understand your concern that the study relies on ten managers from a single organization. In the context of this case, however, these managers collectively supervise approximately 30 projects, representing close to half of the projects active in the organization at the time of data collection (around 70 projects). Because managers typically oversee multiple teams and projects simultaneously, their experiences provide visibility into a substantial portion of the organization’s ongoing development work. Another aspect to consider is that software project managers typically represent a much smaller portion of the workforce in software companies compared to other professionals, such as programmers or testers. As a result, the number of potential participants available for this type of study is naturally more limited. For this reason, we kindly invite the reviewer to consider the number of participants not only in absolute terms but also in light of these contextual factors. Finally, we note that this sample size is also consistent with several qualitative case studies in software engineering that investigate specialized practitioner roles.
>
> Regarding transferability to other contexts, for additional context, our case is a software company founded in 1996 with operational presence in Brazil, Portugal, and the United States. The organization employs approximately 1,200 people, of whom more than 70 percent are involved in software development, organized into over 70 teams delivering projects across sectors such as finance, telecommunications, government, manufacturing, services, and utilities. These teams use a variety of development approaches, including Scrum, Kanban, and Waterfall, and serve clients across several international regions. We will add this information in the paper to allow better assessment of the scope of the case and transferability of results.
>
> Finally, we appreciate your observation that the findings may feel preliminary due to the limited empirical base. In many ways, this aligns with our intention for the study. Given the current scarcity of empirical work examining the experiences of software project managers working with LLM-supported development, our goal is to contribute an exploratory perspective that can serve as a starting point for more extensive investigations in the future.
>
> Thank you again for the constructive feedback and for recognizing the contribution of the managerial perspective explored in this work.

---

### Official Review · Reviewer_KxEN · 2026-03-10

**Rating:** 2
**Confidence:** 3

**Review:**

**Strengths**
- The paper addresses an emerging and important area in software engineering research.
- The study clearly identifies a gap in existing literature, which largely focuses on developer activities rather than managerial adaptation.
- An exploratory case study with thematic analysis is a suitable approach for studying socio-technical phenomena such as managerial perceptions and organizational adaptation.
- The inclusion of participant quotations strengthens credibility and helps illustrate how managers interpret LLM adoption in practice.

**Weaknesses**
- The study includes only 10 participants from a single organization, which limits external validity and transferability.
- The use of an online questionnaire instead of interviews reduces depth and prevents clarification or probing of responses.
- The thematic analysis process is described at a high level but lacks details such as coding schema, codebook examples, or inter-coder agreement metrics.
- Although descriptive statistics are presented, the study does not conduct inferential statistical analysis for quantitative stats.
- The paper claims both qualitative and quantitative data analysis, yet quantitative analysis remains minimal and purely descriptive.
- While the replication package is mentioned, the paper itself does not provide sufficient detail about coding procedures, theme derivation, or data processing to ensure full reproducibility.
- Several sections repeat similar claims (e.g., continuity of managerial skills and incremental learning), which could be aggregated.
- Data rely solely on self-reported perceptions of managers, which may introduce subjective bias and may not reflect actual managerial practices.
- Writeup is rough
- Abrupt sentences and passive voice makes the paper harder to read and understand
- Abstract needs better refinement on what the dataset was
- There is not much stats on how large the org was

The work appears more suitable as an early-stage study rather than a mature empirical contribution. The work requires revisions focusing on methodological transparency, larger dataset, and clearer positioning as qualitative research.

**Summary:**

This paper presents an exploratory case study investigating how software project managers perceive the effects of large language models (LLMs) on software project management practices. The study focuses on managerial perspectives rather than developer-centric views that dominate current research. Data were collected using an online qualitative questionnaire administered to ten project managers working in a large multi-project software organization. Responses were analyzed using thematic analysis and supported by descriptive statistics.

---

> ### Author Response · Authors · 2026-03-13
>
> Dear Reviewer,
>
> Thank you for reading our paper and for the feedback. We appreciate the comments and the opportunity to clarify aspects of the study. To start, we would like to highlight that in conducting and reporting this research, we followed the ACM SIGSOFT Empirical Standards (https://www2.sigsoft.org/EmpiricalStandards/), which define the community’s expectations for empirical software engineering research. Below, we address each limitation raised:
>
> 1. Sample size and single organization: We understand the concern regarding the inclusion of ten participants from a single organization. However, this characteristic is aligned with the nature of case study research, which prioritizes depth of contextual understanding rather than large samples. Case studies investigate a bounded case in detail rather than attempting population level generalization. In qualitative research, there is no universally established number of participants required for a valid study. The appropriate sample size depends on the research context, the specificity of the participant profile, and the depth of the data collected. For instance, Guest (2006) reports that core thematic patterns often emerge after approximately six interviews and that data saturation frequently occurs around twelve interviews, sometimes even earlier. In software engineering research, when the target population is highly specialized, studies may involve even smaller samples. For example, Vogelsang and Borg (2019) conducted a qualitative study on requirements engineering for machine learning with four participants. In our study, the ten participants are software project managers, a relatively small professional group in most organizations. Also, as reported in the paper, these managers collectively supervise approximately 30 projects, representing close to half of the projects active in the organization at the time of data collection (about 70). Therefore, the sample reflects a substantial portion of managerial practice within the case. Finally, the SIGSOFT empirical standards explicitly list criticisms related to small samples or lack of statistical generalization as invalid criticisms for case study research, since the goal is analytical transferability rather than statistical generalization.
>
> References:
> - Guest, G., Bunce, A., & Johnson, L. (2006). How many interviews are enough? An experiment with data saturation and variability. Field methods, 18(1), 59-82
> - Vogelsang, A., & Borg, M. (2019, September). Requirements engineering for machine learning: Perspectives from data scientists. In 2019 IEEE 27th international requirements engineering conference workshops (REW) (pp. 245-251).
>
> 2. Use of an online questionnaire instead of interviews: We agree that interviews could have enabled deeper probing and clarification. We acknowledge this limitation in the paper, and as we said unfortunately, the professional profile of the participants made synchronous interviews impractical. Software project managers reported very limited availability, which made scheduling interviews difficult. In this context, we faced two alternatives: not conducting the study or adapting the instrument to allow participation We chose the second option by transforming the interview guide into an online qualitative questionnaire, which allowed participants to respond asynchronously. While this approach reduces opportunities for probing, it enabled access to a group of practitioners who would otherwise have been unable to participate.
>
> 3. Level of detail in the thematic analysis description: We appreciate this comment. One constraint we faced is the eight-page limit of the conference, which restricts the level of methodological detail that a qualitative study woud request. The paper already describes the main analytical steps: responses were coded iteratively, low-level codes were grouped into broader conceptual categories, and disagreements were resolved through discussion between researchers until a shared interpretation was reached. Regarding inter-coder agreement metrics, the SIGSOFT empirical standards classify such metrics as extraordinary attributes, not essential ones, particularly in exploratory case studies. For this reason, we adopted a commonly used qualitative approach:
> • one researcher conducted the initial coding
> • another researcher reviewed the codes and categories
> • disagreements were resolved through consensus discussion
> This consensus-based approach is widely used in qualitative research. Nevertheless, if necessary, we can expand the description of the coding process and include an image demonstrating the process in the final version.
>
> (continue in the next comment)

---

> > ### Author Response · Authors · 2026-03-13
> >
> > 4. Lack of inferential statistical analysis: Respectfully, this request falls outside the scope of qualitative case study research. Case studies do not aim to produce inferential statistical results. The SIGSOFT empirical standards explicitly state that criticisms such as the absence of statistical analysis do not align with the methodological goals of case study research, which focuses on contextual interpretation rather than statistical inference.
> >
> > 5. Limited quantitative analysis: We agree that the quantitative component of the study is limited. However, this is intentional and aligned with the research design. The quantitative elements are used only to characterize the sample and contextualize the qualitative findings, which is why we rely on descriptive statistics. Given the exploratory nature of the study and the size of the sample, conducting inferential analysis would not be methodologically appropriate and is not part of our goals.
> >
> > 6. Detail of coding procedures and reproducibility: The replication package is provided to increase transparency with the raw collected data. We opted not to include the full coding chain because qualitative analysis inherently involves interpretive judgment by researchers, and we aimed to avoid introducing additional interpretive bias through over-structuring. However, if the reviewer considers this information essential, we can update the replication package to include the full coding process, including: quotation → low level code → high level code → category.
> >
> > 7. Repetition of similar findings: We appreciate this observation. The apparent repetition reflects how participants described their experiences. Many managers independently emphasized similar themes, such as continuity of managerial skills and incremental learning about LLMs. Because the results section reports participants’ experiences, some overlap in themes naturally emerges.
> >
> > 8. Reliance on self-reported perceptions: This characteristic is inherent to qualitative case studies, particularly exploratory studies. The goal of the study is to understand how practitioners interpret and experience a phenomenon in practice. Perception-based data is therefore, an appropriate source of evidence in this type of research. We see this study as an initial empirical step in an area where the literature is still limited ((https://scholar.google.ca/scholar?hl=en&as_sdt=0,5&q=LLMs+AND+%22managers%22+AND+%22software+engineering%22)). In fact, this might be the first study about this conducted using this method. At the end, the findings coming from this case may serve as a foundation for future studies that investigate managerial practices through additional methods or in different organizational contexts.
> >
> > 9. Writeup is rough/Abrupt sentences and passive voice make the paper harder to read and understand: We were somewhat surprised by this comment, as no specific parts of our text were indicated. Nevertheless, we appreciate the feedback and would be glad to revise the manuscript for clarity and readability, if the reviewer could provide us with what parts they request changes, we would be very grateful.
> >
> > 10. Abstract clarity: We agree that the abstract can more clearly describe the dataset and will refine this section accordingly.
> >
> > 11. Organizational context: Thank you for pointing this out. We agree that additional details about the organization would improve the reader’s ability to assess the context of the study. For the reviewer’s reference, the case under study is a large software company founded in 1996 with operational presence in Brazil, Portugal, and the United States. The company develops solutions across multiple sectors, including finance, telecommunications, government, manufacturing, services, and utilities. The organization has approximately 1,200 employees, of which more than 70 percent are directly involved in software development, organized into more than 70 teams. These teams include programmers, quality assurance engineers, designers, and other technical roles, and they work using different development methodologies such as Scrum, Kanban, and Waterfall. The company delivers solutions to clients across North America, Latin America, Europe, and Asia. We will include these contextual details in the final version to strengthen the discussion of the case.
> >
> > Finally, regarding the comment that the work appears more suitable as an early stage study, we note that the paper reports an empirical investigation aligned with exploratory case study research in software engineering, which focuses on analyzing phenomena within real world contexts rather than providing large scale validation. In this sense, the study can serve as a basis for future investigations, particularly given the limited empirical evidence about the experiences of software project managers working with LLMs. As the call for papers welcomes case studies, we hope the work can contribute to discussion within the conference community.

---

### Official Review · Reviewer_Epse · 2026-03-13

**Rating:** 2
**Confidence:** 3

**Review:**

Strengths

- A timely topic

Weakness

- Ten participants from one organization are too narrow. The organization is described only as large and multi-project, with no details, such as headcount, adoption context, which makes it impossible to assess whether the findings transfer to any other setting.

- LLMs increase governance uncertainty, require more human review, and are handled informally. However, these findings are expected. Authors may consider identifying at least one finding that challenges or substantially refines prior assumptions.

- The implications for practice in Section 5 are generic. Statements such as support managers in developing contextual understanding are not actionable. This paper may benefit from providing specific guidance.

- The inter-rater agreement process is described vaguely. The resolution process needs further detail.

- The transferability claim in Section 6 is asserted, not argued. In the current form, authors do not explain how the organizational characteristics shape the findings, or when those findings would and would not apply elsewhere. Without providing such details, the core value from the case study is undermined.

- References [13] and [14] are identical duplicates (Jones 2004). One must be removed and citations corrected throughout.

**Summary:**

The paper studies how software project managers perceive LLM-driven changes to their work. Ten managers at one organization responded to an online questionnaire. Thematic analysis was applied to open-ended responses. The main finding: LLMs shift management toward more judgment-based oversight without introducing new formal practices. The topic is timely. However, the managerial angle is underexplored.

---

> ### Author Response · Authors · 2026-03-13
>
> Dear Reviewer,
>
> Thank you for your careful review. We appreciate the attention to our manuscript. Below, we respond to the weaknesses you raised. Please, note that for this study, we followed the ACM SIGSOFT Empirical Standards, which define the evidence standards (https://www2.sigsoft.org/EmpiricalStandards/) and community expectations for conducting and reporting empirical software engineering research.
>
> 1. Ten participants from one organization are too narrow: We anticipated that comments about the sample size might arise, and would like to clarify the methodological expectations for case study research. Case studies prioritize depth of contextual understanding rather than large samples, as they investigate a bounded case in detail. In qualitative research more broadly, the number of participants depends on the topic under investigation, and there is no universal agreement on a minimum sample size. For example, Guest et al. (2006) report that basic thematic patterns can emerge after approximately six interviews and that data saturation often occurs around twelve interviews, sometimes even earlier. In software engineering research, when the target population is highly specialized, studies may include even fewer participants. As an example, Vogelsang and Borg (2019) conducted a study with four participants in the context of machine learning requirements engineering. Our study focuses specifically on software project managers, a role that represents a small portion of the workforce in most software organizations. As we explain in the paper, managers in this company typically supervise multiple projects simultaneously. In our case, the ten participating managers supervise together approximately 30 projects, which represents nearly half of the projects running in the organization at the time of the study (about 70). For this reason, we invite the reviewer to consider the number of participants within the context of the case rather than as an absolute number. Finally, the ACM SIGSOFT empirical standards explicitly list the following as an invalid criticism of case study research: “Sample of one; findings not generalizable. The point of a case study is to study one thing deeply, not to generalize to a population. Case studies should lead to theoretical generalization.”
>
> References:
> - Guest, G., Bunce, A., & Johnson, L. (2006). How many interviews are enough? An experiment with data saturation and variability. Field methods, 18(1), 59-82
> - Vogelsang, A., & Borg, M. (2019, September). Requirements engineering for machine learning: Perspectives from data scientists. In 2019 IEEE 27th international requirements engineering conference workshops (REW) (pp. 245-251).
>
> 2. The findings appear expected: We understand the reviewer’s interest in identifying findings that challenge prior assumptions. However, qualitative case studies report participants’ interpretations of their work rather than inferring conclusions beyond what is supported by the data. Our goal in this exploratory study was to capture how software project managers describe their experience with LLM usage in practice. For methodological reasons, we cannot introduce interpretations that are not grounded in the participants’ words. Different findings may emerge in other organizational contexts, which is precisely why case studies are valuable: they document how phenomena appear within a specific real-world setting. Because the conference explicitly welcomes case studies, we believe that presenting empirical evidence from this particular context contributes to the discussion of LLM adoption in software engineering practice.
>
> 3. The implications for practice are generic: We appreciate the reviewer’s suggestion to provide more concrete guidance. However, in exploratory case studies, it would be methodologically inappropriate to derive prescriptive guidelines that go beyond the empirical evidence. Exploratory case studies typically aim to identify patterns, document emerging practices and generate insights that motivate future research. At this stage, the evidence we collected allows us to report how managers describe their current practices, but not to formulate prescriptive frameworks or operational guidelines. Developing actionable recommendations would require evidence from additional cases or complementary studies. Indeed, extending this work through broader empirical studies is part of our planned future research. It is also important to note that empirical studies specifically examining LLM usage by software project managers are still scarce (https://scholar.google.ca/scholar?hl=en&as_sdt=0,5&q=LLMs+AND+%22managers%22+AND+%22software+engineering%22). Because of this limited body of evidence, our exploratory findings aim to contribute initial empirical observations that may support the development of future research.
>
> (continue in the next comment)

---

> > ### Author Response · Authors · 2026-03-13
> >
> > 4. The inter-rater agreement process is described vaguely: We appreciate this comment and agree that the description can be clarified. Yet, according to the SIGSOFT empirical standards, inter-rater reliability is considered an extraordinary attribute, not an essential one, particularly in exploratory case studies. For this reason, we did not compute a statistical reliability metric. Instead, we followed a commonly used qualitative research practice in which one researcher conducted the initial coding, another researcher reviewed the codes and categories, and disagreements were discussed in consensus meetings until a shared interpretation was reached. This consensus-based coding process is widely used in qualitative studies across multiple disciplines. Nevertheless, if our description in Section 3.5 is not sufficiently clear, we will gladly expand it in the final version.
> >
> > 5. Transferability is asserted rather than argued: We agree with the reviewer that additional contextual details would strengthen the discussion of transferability. For the reviewer’s reference, the case under study is a large software company founded in 1996 with operational presence in Brazil, Portugal, and the United States. The company develops solutions across multiple sectors, including finance, telecommunications, government, manufacturing, services, and utilities. The organization has approximately 1,200 employees, of which more than 70 percent are directly involved in software development, organized into more than 70 teams. These teams include programmers, quality assurance engineers, designers, and other technical roles, and they work using different development methodologies such as Scrum, Kanban, and Waterfall. The company delivers solutions to clients across North America, Latin America, Europe, and Asia. We agree that including these details in the paper will improve readers’ ability to assess the transferability of the findings. If the paper is accepted, we will incorporate this information and strengthen the discussion of contextual conditions under which the findings may or may not apply.
> >
> > 6. Duplicate references: Thank you for pointing out the duplicate reference (Jones 2004). We will remove the duplicated entry and correct the citations throughout the manuscript.
> >
> > Once again, we thank the reviewer for the thoughtful comments. We believe these suggestions will help improve the clarity and contextualization of the final version of the paper.